# Induction of Broadly Cross-Reactive Antibodies by Displaying Receptor Binding Domains of SARS-CoV-2 on Virus-like Particles

**DOI:** 10.3390/vaccines10020307

**Published:** 2022-02-16

**Authors:** Xinyue Chang, Xuelan Liu, Mona O. Mohsen, Andris Zeltins, Byron Martina, Monique Vogel, Martin F. Bachmann

**Affiliations:** 1Department of Rheumatology and Immunology, University Hospital Bern, 3010 Bern, Switzerland; xinyue.chang@dbmr.unibe.ch (X.C.); mona.mohsen@dbmr.unibe.ch (M.O.M.); monique.vogel@dbmr.unibe.ch (M.V.); 2Department of BioMedical Research, University of Bern, 3012 Bern, Switzerland; 3International Immunology Centre, Anhui Agricultural University, Hefei 230036, China; xuelan.liu@dbmr.unibe.ch; 4Saiba GmbH, 8088 Pfäffikon, Switzerland; 5Latvian Biomedical Research & Study Center, Ratsupites 1, LV1067 Riga, Latvia; anze@biomed.lu.lv; 6Artemis Bio-Support, 2629 Delft, The Netherlands; b.martina@artemis-biosupport.com; 7Jenner Institute, University of Oxford, Old Road Campus, Roosevelt Drive, Oxford OX3 7BN, UK

**Keywords:** COVID-19, vaccine, virus-like particle, CuMV_TT_

## Abstract

The impact of the COVID-19 pandemic has been reduced since the application of vaccination programs, mostly shown in the reduction of hospitalized patients. However, the emerging variants, in particular Omicron, have caused a steep increase in the number of infections; this increase is, nevertheless, not matched by an increase in hospitalization. Therefore, a vaccine that induces cross-reactive antibodies against most or all variants is a potential solution for the issue of emerging new variants. Here, we present a vaccine candidate which displays receptor-binding domain (RBD) of SARS-CoV-2 on virus-like particles (VLP) that, in mice, not only induce strong antibody responses against RBD but also bind RBDs from other variants of concern (VOCs). The antibodies induced by wild-type (wt) RBD displayed on immunologically optimized Cucumber mosaic virus incorporated tetanus toxin (CuMV_TT_) VLPs bind to wt as well as RBDs of VOCs with high avidities, indicating induction of strongly cross-reactive IgG antibodies. Interestingly, similar cross-reactive IgA antibodies were induced in immunized mice. Furthermore, these cross-reactive antibodies demonstrated efficacy in neutralizing wt (Wuhan) as well as SARS-CoV-2 VOCs (Beta, Delta, and Gamma). In summary, RBDs displayed on VLPs are capable of inducing protective cross-reactive IgG and IgA antibodies in mice, indicating that it may be possible to cover emerging VOCs with a single vaccine based on wt RBD.

## 1. Introduction

The COVID-19 pandemic recently has been more dominated by variants of concern (VOCs) that have emerged in different countries around the globe [1,2,3,4,5]. Mutations in the RBDs of these VOCs have been characterized in detail, and it was found that some mutations (e.g., E484K present in B.1.351 and P.1 variants) reduce recognition by antibodies induced by the wt SARS-CoV-2 strain [6,7,8], while other mutations (N501Y in B.1.1.7, B.1.351 and P.1 variants or L452R and E484Q in B.1.617.1 variant) primarily enhance affinity for the ACE2 receptor (Table 1) [9,10] likely being responsible for the enhanced infectivity of the latter strains. Both types of mutations cause reduced neutralization of VOCs by wt SARS-CoV-2 induced convalescent sera, either due to reduced recognition of the RBD or impaired competition of RBD with its receptor ACE2 [9].

While recognition by convalescent sera of some RBDs of VOC, e.g., E484K, is essentially abolished, vaccine-induced antibodies seem to be more cross-reactive in the fact that they recognize the newly emerged RBDs with enhanced avidity compared to convalescent sera and show some cross-neutralization both in humans [6] and mice [12]. While attempts have been made to modify mRNA vaccines to adjust for the new VOCs, success was rather limited, and neutralization of VOCs induced by the adjusted vaccine was not better than the values achieved with the wt mRNA vaccine [13]. Hence, a simpler strategy may be to search for vaccine candidates that induce neutralizing antibodies against all VOCs.

CuMV_TT_ is a previously described virus-like particle (VLP) derived from the cucumber mosaic plant virus, which has been immunologically optimized by incorporating a universal T helper cell epitope derived from tetanus toxin, expected to facilitate induction of T helper cell-dependent IgG responses [14,15]. In addition, CuMV_TT_ is expressed in E. coli and spontaneously packages bacterial RNA, which serves as a TLR7/8 ligand and is optimal for enhancing IgG and IgA responses [16]. For example, CuMV_TT_-Ara h 1 was demonstrated to effectively treat peanut allergy in mice [17]. Another example of CuMV_TT_ as an excellent vaccine delivery vehicle was against MERS coronavirus, which was potent to induce high titers of RBD- and spike protein-specific IgG antibodies, which could neutralize virus in vitro [18].

Here we demonstrate that wt RBD displayed on CuMV_TT_ may be a candidate for inducing polyvalent protective immunity, as it induces antibodies in mice that broadly recognize and neutralize all VOCs tested.

## 2. Materials and Methods

### 2.1. Animal Ethics

All studies involving animals were approved by the Cantonal Veterinary Office in Bern, Switzerland. All animal experiments were performed according to regulations and guidelines of the Cantonal Veterinary Office Bern, Switzerland (license BE70/18). There is no human material involved in this study.

### 2.2. RBD Proteins Expression and Purification

The production and purification of RBD proteins were described previously [19]. Briefly, The RBD variants (RBD_wt_, RBD_trip_, RBD_417_, RBD_484_, RBD_501_, and RBD_dm_) were expressed in mammalian Expi293F cells (Gibco, ThermoFisher Scientific, Waltham, MA, USA). Plasmids containing genes encoding wild type and mutated RBD proteins (residues Arg319-Phe541) with a 6-His tag at C-terminus were synthesized from Twist Bioscience, San Francisco, USA and amplified in bacterial XL-1 Blue competent cells. Then Expi293F cells (3 × 10^6^ cells/mL) were transfected with expression plasmids by using ExpiFectamine 293 transfection kit (Gibco, Thermo Fisher Scientific, Waltham, MA, USA) according to the manual. Seven days after transfection, cell culture supernatants containing RBDs were collected by centrifugation and filtration. RBDs were purified with His-Trap HP column (GE Healthcare, Wauwatosa, WI, USA) and dialyzed in PBS buffer. Purified RBDs were analyzed in SDS-PAGE gel.

### 2.3. Displaying RBDs on VLP Surface

RBD_wt_ protein was covalently linked to CuMV_TT_ VLPs with the cross-linker Succinimidyl 6-(beta-maleimidopropionamido) hexanoate (SMPH) (Thermo Fisher Scientific, Waltham, MA, USA), as described previously [19]. The linker firstly was incubated with CuMV_TT_ for 30 min at room temperature, and meanwhile, RBD_wt_ was reduced by mild reducing agent Tris-(2-Carboxyethyl) phosphine (TCEP) (Invitrogen, Waltham, MA, USA). The molar ratio of SMPH to CuMV_TT_ was optimized to 5:1, and RBD_wt_ to CuMV_TT_ was 1:1. Then the CuMV_TT_ was mixed with reduced RBD_wt_ protein and incubated for 3 h with 400 rpm shaking. Subsequently, the coupling was analyzed in SDS-PAGE gel, and coupling efficiency was calculated by densitometry (20–30% efficiency). In addition, the conjugated CuMV_TT_-RBD_wt_ products were characterized in agarose gel to determine the content of ssRNA.

### 2.4. Transmission Electron Microscope (TEM)

The integrity of the CuMV_TT_-RBD_wt_ VLP sample was assessed by TEM, and 5 µL of VLP solutions were dropped on grids for 1 min and rinsed by dipping into the water 3 times. The grids were then stained with 2% uranyl acetate solution (Electron Microscopy Science, Hatfield, PA, USA) for 45 s. The excess solution was gently removed by filter paper. Images were captured with a digital camera (Veleta, Olympus, Münster, Germany) under a transmission electron microscope (Tecnal Spirit, FEI, Hillsboro, OR, USA) at 80 kV.

### 2.5. Immunization of Mice

*BALB/cOlaHsd* mice (female) were purchased from Envigo (Horst, The Netherlands) at the age of 7 weeks. Mice were kept in a specific pathogen-free (SPF) facility in the Department of BioMedical Research (DBMR) of the University of Bern, Switzerland, according to the guidelines of Cantonal Veterinary. All animal experiments were performed according to ethical principles and guidelines of Cantonal Veterinary Office Bern, Switzerland. Five Female *BALB/c* mice (8–12 weeks) were subcutaneously immunized with 40 µg CuMV_TT_-RBD_wt_ at d0 and boosted at d28. Serum samples were collected at d14, d21, d28, d35, d42, and d49 after priming. All mice in experiments did not show body weight loss and survived until d49, when we euthanized the mice.

### 2.6. ELISA

Antibody responses in immunized mice were examined with ELISA. Shortly, half-well Corning 96-well plate was coated with RBDs (1 µg/mL) at 4 °C overnight. Then the plate was blocked at room temperature for 2 h with PBS-0.15% Casein. Afterwards, serum samples were added, and a 3-fold serial dilution from 1:40 was performed. After 1 h incubation at room temperature, goat anti-mouse IgG-HRP (Jackson ImmunoResearch, West Grove, PA, USA) was added and incubated for 1 h. Finally, the developing solution (TMB in citrate buffer) was added, and the stop solution (1 M H_2_SO_4_) was added 5 min later. The plate was read at OD_450nm,_ and IgG titer was calculated as the serum dilution times that reach half the maximum OD value.

To determine RBD-specific IgA antibodies in serum samples, the same assays were performed with some adjustments. Serum samples were incubated with Protein G magnetic beads (Thermo Scientific, Waltham, MA, USA) for 10 min at room temperature to remove IgG antibodies. Instead of an anti-IgG antibody, goat anti-mouse IgA-HRP antibody (ICN Cappel, Costa Mesa, CA, USA) was used as a detecting antibody.

### 2.7. Avidity ELISA

To assess the quality of RBD-specific IgG antibodies induced by CuMV_TT_-RBD_wt_, two parallel plates were performed as described in Section 2.6. To distinguish high-avidity binding IgG antibodies from low-avidity binding ones, one plate was washed 3 times for 5 min with 7 M urea and the other with PBS-0.05% Tween [20]. The amounts of high-avidity antibodies were displayed with titer as well.

### 2.8. Neutralization Assay

The ability of immunized serum samples to neutralize SARS-CoV-2 virus Wuhan, South Africa (SA), Delta, and Brazilian variants were tested by means of cytopathic effect (CPE) formation [19]. Firstly, serum samples were heat-inactivated at 56 °C for 30 min and then 2-fold diluted from 1:20 until 1:160. Then, 100 TCID_50_ of Wuhan wild type (SARS-CoV-2/ABS/NL20), SA (Beta: SARS-CoV-2/ABSB/BL21), Delta (SARS-CoV-2/ABSD/NL21), and Brazilian variant virus were added to each diluted serum and incubated for 1 h at 37 °C. Subsequently, the mixture was added to a monolayer of Vero cells and incubated for 4 days at 37 °C. Finally, the wells were inspected for the presence of cytopathic effect (CPE). Titer was expressed as the highest dilution of the serum that fully inhibits the formation of 100% CPE.

### 2.9. Statistical Analysis

Statistical analysis was performed using GraphPad Prism 7.0d. Unpaired Student’ *t*-tests were used for IgG titers in ELISA, differences between plates with and without urea wash in avidity ELISA assays and neutralization titers. Statistical significance was indicated as *p* values: ≤0.05 (*), ≤0.01 (**), ≤0.001 (***), ≤0.0001 (****).

## 3. Results

### 3.1. Displaying RBD_wt_ on CuMV_TT_ VLP Surfaces

The production of the CuMV_TT_-RBD_wt_ vaccine has been described previously [19]. In brief, we chemically coupled RBD_wt_ protein expressed in eukaryotic HEK293 cells to CuMV_TT_ VLPs. Figure 1A illustrates the principles of covalent reactions, which links the RBD_wt_ internal cysteine to a lysine on the surface of the CuMV_TT_ subunit via the heterobifunctional cross-linker SMPH. Efficient conjugation was confirmed by SDS-PAGE analysis, where a single VLP subunit can be distinguished from subunits coupled to RBD_wt_. In Figure 1B, the CuMV_TT_-RBD_wt_ conjugation band is marked with a star. Because CuMV_TT_ VLPs are produced in *E. coli*, they spontaneously package prokaryotic ssRNA to stabilize the capsid structure. As shown in agarose gel stained for RNA (Figure 1C), CuMV_TT_-RBD_wt,_ as well as unmodified CuMV_TT_ packaged ssRNA (left panel). The same gel stained for protein showed a band at the corresponding positions (right panels, Coomassie staining), indicating the presence of intact VLPs. Furthermore, the shape of CuMV_TT_-RBD_wt_ was examined by transmission electron microscopy (Figure 1D), which demonstrated that CuMV_TT_-RBD_wt_ remained as intact viral particles. These results demonstrated that the CuMV_TT_-RBD_wt_ vaccine kept the properties of parent CuMV_TT_ VLPs that incorporated RNA and preserved a spherical shape.

### 3.2. CuMV_TT_-RBD_wt_ Induces Strong Cross-Reactive IgG Responses

The immunogenicity of CuMV_TT_-RBD_wt_ has been illustrated in previous work for wt RBD [19]. Here we assessed whether the antibodies are reactive to the newly emerged SARS-CoV-2 variants of concern (VOCs), which showed increased transmissibility, for instance, B.1.1.7, B.1.351, P.1 and B.1.617.1 variant. Of note, mutations in RBD protein, for example K417N and E484K, present in B.1.351 and P.1 variants reduce recognition by convalescent sera, while N501Y in B.1.1.7, B.1.351, and P.1 variants, L452R and E484Q in B.1.617.1 variant, increased the affinity to ACE2 receptor thereby causing reduced neutralization by affinity-escape [21,22] (Table 1). To this end, we produced recombinant RBD proteins containing the above mutations, namely RBD_417_, RBD_484_, RBD_501_, RBD_trip,_ which includes three K417N, E484K, and N501Y mutations, and RBD_dm_ containing L452R and E484Q mutation (shown as Appendix A). Mice were immunized with CuMV_TT_-RBD_wt_ at d0 and boosted at d28, and serum samples were collected weekly until d49, as illustrated in Figure 2A. ELISA results showed that antibodies induced by CuMV_TT_-RBD_wt_ were able to recognize not only RBD_wt_ but also all other mutated RBDs (Figure 2B–G). Overall, IgG antibody titers against all RBDs increased with time, especially after the booster injection.

IgG titers against RBD_wt_ were superior to some of the RBD mutants after the first injection. After the second injection, differences, however, levelled out. Interestingly, as observed with convalescent sera and sera from immunized individuals (mRNA) [6], IgG titers against RBD_484_ and RBD_trip_ were significantly reduced after a single injection. In contrast, the titers against RBD_417_, RBD_501_, and RBD_dm_ were only slightly lower, again reflecting what has been observed with convalescent sera and sera from immunized individuals (Figure 2B). In addition, the difference between IgG titers against RBD_wt_, RBD_trip_, and RBD_dm_ were significant before the booster (Figure 2C,D). These results suggest that the CuMV_TT_-RBD_wt_ vaccine was able to induce IgG antibodies that cross-react with VOCs, in particular after boosting; the ability to recognize RBD_484_, RBD_trip_, and RBD_dm_ was the lowest, mirroring experience in humans with natural infection and mRNA immunization.

The differences between titers against RBD_wt_ and RBD mutants became, however, much less pronounced, and all IgG titers were elevated to similar levels after the booster, albeit titers against RBD_wt_ stayed highest (Figure 2E–G). Thus, cross-reactive IgG antibodies reached similar levels after the booster, indicating the booster vaccination unified the IgG responses.

### 3.3. CuMV_TT_-RBD_wt_ Induces Strong Cross-Reactive IgA Responses

The ability of CuMV_TT_-RBD_wt_ to induce cross-reactive IgA antibodies was assessed next. We have shown previously that RNA packaged in VLPs may drive systemic IgA responses [23], and IgA is well known to serve as a noninflammatory isotype that protects mucosal surfaces against infection. To this end, sera from immunized mice were tested for the presence of IgA antibodies specific to the various RBDs on d14 and d49 after immunization. As shown in Figure 3, IgA responses remained low after the first injection but strongly increased after the booster and reached high levels by day 49. As seen for the IgG responses, the IgA responses were highly cross-reactive, with mutation E484K being least recognized.

### 3.4. CuMV_TT_-RBD_wt_ Induces IgG Antibodies Recognizing VOCs with High Avidity

To assess the avidity of the induced antibodies, we performed a modified immunoassay using 7M urea to detach low avidity antibodies. More specifically, we compared the avidity of the antibodies against the various RBDs in sera collected from day 14 and day 49. As expected, the avidity of antibodies on day 14 was relatively low for all RBDs, indicated by the strongly reduced ELISA signals in the plates washed with 7M urea (Figure 4A, red symbols). In contrast, avidity was strongly increased on day 49 after the boost. Remarkably, the observed avidity was similar on all RBDs of VOCs and almost identical to avidities measured against wt RBD (Figure 4B).

### 3.5. CuMV_TT_-RBD_wt_ Induces IgG Antibodies Broadly Cross-Neutralizing VOCs

To determine the ability of the induced antibodies to neutralize VOCs, we have assessed the reduction of cytopathic effect (CPE) using 100TCID50 of wild type and SARS-CoV-2 variants. Titers are expressed as the highest dilution that inhibits 100% formation of CPE. Interestingly, the d49 sera neutralized Wuhan (wt) as well as the Beta and Delta mutant at high titers and to the same extent. Only the Gamma variant was reduced to some degree (Figure 5). The reduced neutralization of the Gamma variant may be due to subtle differences in RBD (Table 1) or inherent differences between the viruses contributing to in vitro neutralization. Hence, these data demonstrate the ability of wt RBD displayed on CuMV_TT_ to induce highly cross-reactive and cross-neutralizing antibodies, indicating that yearly generation of adapted vaccine generations as known for flu may not be necessary.

## 4. Discussion

The recently emerging VOCs raised the question of whether the currently available and developed vaccines may require yearly adaptation, as is known from the influenza virus [24]. This may not be the case for SARS-CoV-2 due to several reasons. (1) While both SARS-CoV-2 and influenza virus are RNA viruses, SARS-CoV-2 has a much more stable genome due to the presence of a proof-reading system [25]. (2) SARS-CoV-2 has a highly defined receptor, namely ACE2 and the binding is based on protein–protein interaction; indeed, the receptor-binding motif which is directly responsible for interaction with ACE2 shows no glycosylation or other post-translational modifications [26]; altered post-translational modifications are therefore not likely to interfere with this protein-protein interaction. In contrast, the influenza virus binds to variable carbohydrate structures via hemagglutinin and therefore has much more room for modifications [27,28]. (3) Consistent with (2), the independently emerged VOCs share a lot of common mutations, with position E484 or N417 or K501 being very frequently mutated. This also demonstrates that the virus has a limited operating range in terms of mutations. Nevertheless, the recently emerged Omicron VOC shows a larger number of mutations, and it will be interesting to reveal how this variant behaves in terms of neutralization. (4) The virus recently jumped from animals to humans, and it is therefore likely that these mutations observed now are part of the process of “human adaptation”. This theory is supported by the finding that the mutation within RBD, which determines the binding affinity of RBD to ACE2 and replication in human lung cells, is a key driver of SARS-CoV-2 to human adaptation [29]. (5) Established RNA viruses, such as Polio or Dengue virus, exist in three or four serotypes, respectively and show no evidence of further mutations to escape immunity. Indeed, poliovirus has been held at bay for decades using the same vaccine [30]. (6) Finally, we show here that a vaccine based on wt RBD displayed on CuMV_TT_ is able to induce strongly and broadly cross-protective antibodies recognizing all VOCs efficiently. Taking points 1–6 into account, it will likely not be necessary to generate yearly versions of a SARS-CoV-2 vaccine. In addition to vaccine optimization, the world should therefore focus on broadening vaccine accessibility rather than vaccine specificity.

## 5. Conclusions

Here, we present a vaccine candidate against Wuhan SARS-CoV-2 based on virus-like particle, CuMV_TT_-RBD_wt_, which can induce antibodies recognize not only RBD_wt_, but also RBDs of VOCs. These antibodies are able to neutralize VOCs in vitro as well. These results may be support the rationale for administration of wt vaccines globally to prevent the VOCs.

## Figures and Tables

**Figure 1 vaccines-10-00307-f001:**
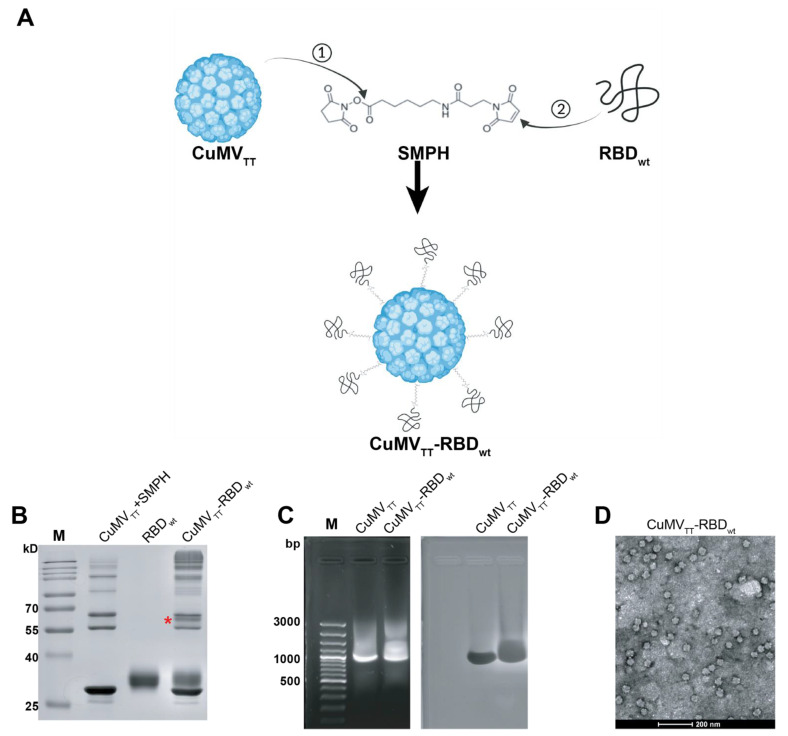
Generation and characterization of CuMV_TT_-RBD_wt_. (**A**) Scheme of covalent reactions between CuMV_TT_, RBD_wt_ and SMPH. (**B**) SDS-PAGE gel analysis of CuMV_TT_-RBD_wt_. The coupled CuMV_TT_-RBD_wt_ band (56.3kD) was marked with a star. (**C**) Examination of RNA contents inside CuMV_TT_-RBD_wt_ comparing with CuMV_TT_ VLP (left); the same agarose gel was stained with Coomassie, illustrating the protein bands located at the same position as RNA bands (right). (**D**) Transmission electron microscope image of CuMV_TT_-RBD_wt_, which demonstrates sphere viral shape.

**Figure 2 vaccines-10-00307-f002:**
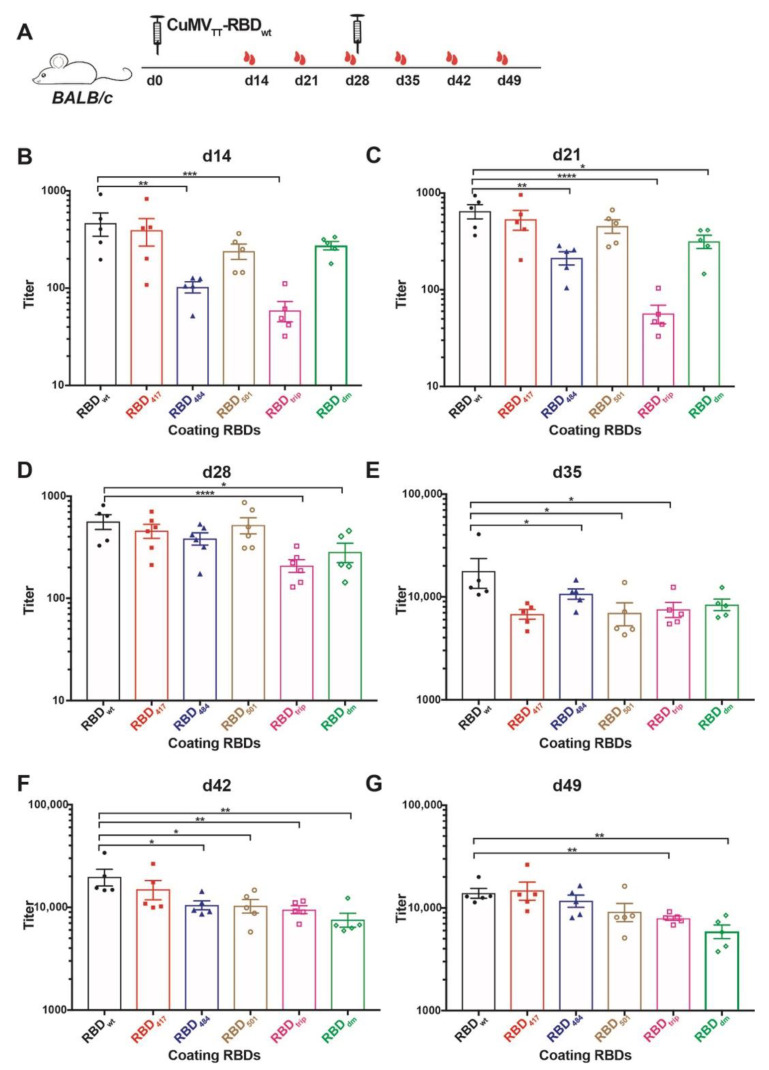
Assessment of immunogenicity of CuMV_TT_-RBD_wt_ in mice to RBDs of VOCs. (**A**) *BALB/c* mice were immunized with 40 µg CuMV_TT_-RBD_wt_ at day 0 and boosted at day 28. Serum samples were collected weekly until d49. (**B**–**G**) IgG titers of d14 (**B**), d21 (**C**), d28 (**D**), d35 (**E**), d42 (**F**), d49 (**G**) against RBDwt, K417N mutation RBD_417_, E484K mutation RBD_484_, N501Y mutation RBD_501_, K417N, E484K, and N501Y mutation RBD_trip_, and L452R, E484Q mutation RBD_dm_. Statistical significance was analyzed using unpaired t-test, *p* values: ≤0.05 (*), ≤0.01 (**), ≤0.001 (***), ≤0.0001 (****).

**Figure 3 vaccines-10-00307-f003:**
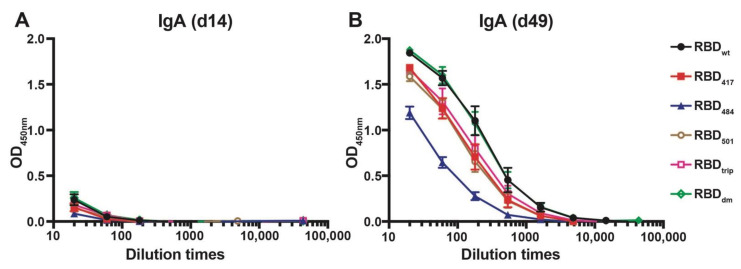
IgA antibody responses against all RBDs before boost (d14, (**A**)) and after boost (d49, (**B**)) of mice immunized with CuMV_TT_-RBD_wt_.

**Figure 4 vaccines-10-00307-f004:**
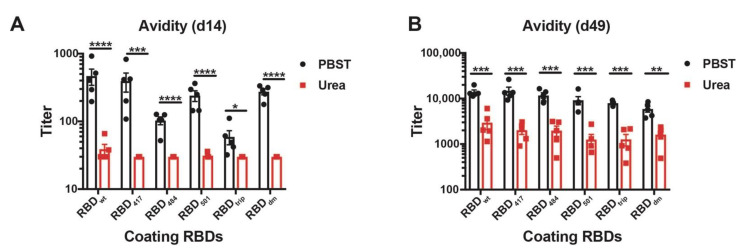
Avidity ELISA of CuMV_TT_-RBD_wt_ immunized mice sera to all RBDs at d14 (**A**) and d49 (**B**). IgG titers with (red symbols) and without (black symbols) 7M urea wash were displayed. Statistical significance was analyzed using unpaired t-test, *p* values: ≤0.05 (*), ≤0.01 (**), ≤0.001 (***), ≤0.0001 (****).

**Figure 5 vaccines-10-00307-f005:**
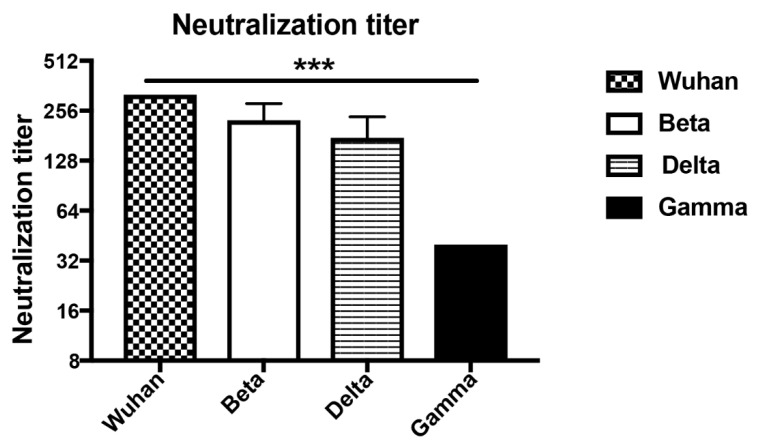
Titers of day 49 sera after immunization neutralizing Wuhan, Beta (South Africa), Delta (Indian) and Gamma (Brazilian) SARS-CoV-2 strains. Neutralization titers were determined as the highest dilution to inhibit 100% CPE formation. Statistical significance was analyzed using unpaired t-test, *p* values: ≤0.001 (***).

**Table 1 vaccines-10-00307-t001:** Mutations of VOCs compared with wild type in RBD [11] and corresponding affinity to ACE2 [9,10].

Strain	First Reported Country	Mutations in RBD	Affinity to ACE2
Wild type	Wuhan	K417, L452, T478, E484, N501	20.5 × 10^−9^ M
Alpha (B.1.1.7)	United Kingdom	N501Y	6.2 × 10^−9^ M
Beta (B.1.351)	South Africa	K417N, E484K, N501Y	10.3 × 10^−9^ M
Gamma (P.1)	Brazil	K417T, E484K, N501Y	/
Kappa (B.1.617.1)	India	L452R, E484Q	4.6 × 10^−9^ M

## Data Availability

All data generated or analyzed during this study are included in this published article (and its Appendix A).

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
