# Peer review of "Induction of Broadly Cross-Reactive Antibodies by Displaying Receptor Binding Domains of SARS-CoV-2 on Virus-like Particles"

_vaccines, 2022, doi:10.3390/vaccines10020307_

Round 1

Reviewer 1 Report

In this manuscript, Martin F. Bachmann and his colleagues displayed SARS-CoV-2 RBD on CuMVtt VLPs surfaces and then validated its ability to induce IgG and IgA responses in mice. The design of the study, in general, is reasonable. However, the presentation of the study is far from satisfactory. Firstly, the background introduction is too general and vague. It did not provide enough precise information, such as a lack of references. Secondly, the authors failed to discuss this study surrounding their findings and, unfortunately, lost the focus on the discussion part.  

Comments:

  1. Please show the full name of CuMVtt in line 21.
  2. References are insufficient. For instance, there are no references from line 32 to line 37 to support these descriptions.
  3. The summary in table 1 is not complete. It would be best if you also mentioned Delta and Omicron.
  4. Table 1 is in the wrong place. You can not show it in the middle of the introduction part.
  5. The description of figure 2A is confusing. Based on my understanding, figure 2 tested different RBDs. If it labels CuMVtt-RBDwt in figure 2A, it means the authors only tested WT. Therefore, figure 2A should mark CuMVtt-RBD rather than CuMVtt-RBDwt. Also, in line 179, there is the same mistake.
  6. In line 192, for the sentence “as observed with convalescent sera and sera from immunized individuals (mRNA),” which paper did you refer to?
  7. In line 235, where are the d49 sera from?
  8. In lines 235-237, why was the neutralization ability reduced in the Brazilian strain? It would be helpful to compare the RBD differences among tested strains and give some explanations.

Author Response

Comment 1: Please show the full name of CuMVtt in line 21.

Reply: we have added the full name of CuMVTT in revised text.

Comment 2: References are insufficient. For instance, there are no references from line 32 to line 37 to support these descriptions.

Reply: we have added the references accordingly.

Comment 3: The summary in table 1 is not complete. It would be best if you also mentioned Delta and Omicron.

Reply: The reason we did not mention was that there was no in vitro affinity of Delta and Omicron published, but in silicon studies were available. Recently, there is one study measured affinity of RBDs of WT, Delta and Omicron to ACE2 by non-competitive ELISA, which are 16 ´ 10-9 M, 3.7 ´ 10-9 M and 270 ´ 10-9 M (Wu, L., Zhou, L., Mo, M. et al. SARS-CoV-2 Omicron RBD shows weaker binding affinity than the currently dominant Delta variant to human ACE2. Sig Transduct Target Ther 7, 8 (2022). https://doi.org/10.1038/s41392-021-00863-2). However, these numbers are calculated from different methods, we decided to not include them.

Comment 4: Table 1 is in the wrong place. You can not show it in the middle of the introduction part.

Reply: The layout was performed by editor, we leave that to the editor.

Comment 5: The description of figure 2A is confusing. Based on my understanding, figure 2 tested different RBDs. If it labels CuMVtt-RBDwt in figure 2A, it means the authors only tested WT. Therefore, figure 2A should mark CuMVtt-RBD rather than CuMVtt-RBDwt. Also, in line 179, there is the same mistake.

Reply: We tested the CuMVTT-RBDwt elicited sera to recognize different RBDs. The CuMVTT-RBDwt in figure 2A was to demonstrate the vaccine was wild type RBD. We added more labeling to make it more understandable.

Comment 6: In line 192, for the sentence “as observed with convalescent sera and sera from immunized individuals (mRNA),” which paper did you refer to?

Reply: Sorry for the careless neglect. We have added the reference in revised text.

Comment 7: In line 235, where are the d49 sera from?

Reply: The sera are from mice immunized with CuMVTT-RBDwt, day 49 after prime immunization.

Comment 8: In lines 235-237, why was the neutralization ability reduced in the Brazilian strain? It would be helpful to compare the RBD differences among tested strains and give some explanations.

Reply: The sequence difference of RBD of South African and Brazilian varients was K417N or K417T, respectively. That may have an influence. On the other hand, the viruses may have intrinsically different susceptibilities to neutralizing antibodies.

Reviewer 2 Report

This study introduced a vaccine to induce cross-reactive antibodies against SARS-CoV-2 VOCs. The results shown the effectiveness of this method. It may be useful for vaccine development. In general, the design of this study is good. Here are some suggestions may help to improve the quality of this manuscript.

  1. Line 16: “Abstract: The COVID-19 pandemic has slowed down since the application of vaccination.” Since the outbreak of SARS-Cov-2 Omicron variant. The cases of covid-19 increasing again. Please rewrite accordingly.
  2. Please notice SARS-Cov-2 VOCs: South Africa is Beta variant and Brazilian is Gamma variant.
  3. Authors should introduce more about CuMVTT in the introduction, including other works based on CuMVTT and the advantage of CuMVTT system compare with regular vaccine.
  4. Line 111 and line 119. I can’t find the “goat anti-mouse IgG-POX” and “goat anti-mouse IgA-POX” antibodies at Jackson ImmunoResearch. According to your previous publication, POX should be HRP. Please use HRP instead.
  5. Line 142, 189 and 230. in p values, both *** and **** were ≤ 0.001?
  6. Since the MW of CuMVTT-RBDWT> 55 KD, please add the size of the ladders bigger than 55KD In Figure 1B.
  7. Please add the mouse data into this manuscript including body weight and survival.

Author Response

Comment 1: Line 16: “Abstract: The COVID-19 pandemic has slowed down since the application of vaccination.” Since the outbreak of SARS-Cov-2 Omicron variant. The cases of covid-19 increasing again. Please rewrite accordingly.

Reply: We have changed the text to “The COVID-19 pandemic has been improved since the application of vaccination shown as the reduction of hospitalized patients. However, the emerging variants Omicron causes a steep increase in infections; this increase in infection, however, is apparently not matched by increased hospitalizations.”

Comment 2: Please notice SARS-Cov-2 VOCs: South Africa is Beta variant and Brazilian is Gamma variant.

Reply: This is now consistent with the information in Table 1.

Comment 3: Authors should introduce more about CuMVTT in the introduction, including other works based on CuMVTT and the advantage of CuMVTT system compare with regular vaccine.

Reply: We have added some examples in the introduction about CuMVTT.

Comment 4: Line 111 and line 119. I can’t find the “goat anti-mouse IgG-POX” and “goat anti-mouse IgA-POX” antibodies at Jackson ImmunoResearch. According to your previous publication, POX should be HRP. Please use HRP instead.

Reply: We have changed the text accordingly.

Comment 5: Line 142, 189 and 230. in p values, both *** and **** were ≤ 0.001?

Reply: Thank you for pointing this careless mistake out. We have changed the text.

Comment 6: Since the MW of CuMVTT-RBDWT> 55 KD, please add the size of the ladders bigger than 55KD In Figure 1B.

Reply: We have added the size of ladder in figure 1B.

Comment 7: Please add the mouse data into this manuscript including body weight and survival.

Reply: We have added the information in Section 2.5.

Reviewer 3 Report

This is an interesting paper where the authors demonstrate the capability of a novel CuMVTT-based vaccine in inducing IgA and IgG Abs in a mouse model. While the methodology and results are solid and the findings are of interest, the manuscript does not adequately reflect study findings in the context of previous study findings examining candidate SARS-CoV-2 vaccines and cross-reactive Abs. For instance, the discussion mainly outlines other viral models (Polio, Dengue) previous SARS-CoV-2 variants, exclusively focusing on Ab responses, with the authors concluding that “it will not be necessary to generate yearly versions of the SARS-CoV-2 vaccine”. This conclusion is just too simplistic a conclusion for a small study conducted in mice. Suggest making inferences not based on study findings.

Comments for the authors are outlined below:

Abstract

Page 1 – The abstract is a little misleading. It suggests that the cross-reactive IgG Ab experiments were not performed in mice as only the IgA Ab experiments were reported as being conducted in mice (leading to the assumption that these experiments were conducted in another animal model or humans). Suggest clarifying that all experiments were in mice (discovered after reading the methods). Therefore, suggest the statement  ‘VLPs are capable of inducing protective cross-reactive IgG and IgA Abs’ should have “in mice” added.

Introduction

Page 1, line 35 – need to include a reference to that particular variant enhance affinity for the ACE2 receptor and infectivity of latter strains.

Page 2, line 51 – suggest describing / expanding that CuMVTT is a vaccine to the reader who may not be familiar with this field.

Methods

It is not clear how many mice were in each group. It appears from Fig 2 to be 4-5. Suggest outlining sample size of each group in the methods.

Results

Figure 5 - The authors are to be commended for examining the cross-neutralising ability of induced Abs in comparison to just examining seroconversion. However, it would be more appropriate to label variants as alpha, beta, delta, gamma, etc for consistency and clarity.

Would the authors consider including a figure comparing the kinetics of IgG versus IgA induction?

Discussion

Page 9, line 245 – The discussion (one large paragraph) focuses predominantly on whether it will be necessary to generate yearly versions of the SARS-CoV-2 vaccine. The standard format for a discussion is to provide a  summary of the overall study findings in the first paragraph (there is a only brief outline of the results in point 6 but no specific mention of IgG and IgA), then discuss these results with respect to other study findings and clinical implications of the results.

While the one paragraph is of interest discussing whether it will be necessary to generate yearly versions of the SARS-CoV-2 vaccine, eliciting Ab responses to SARS-CoV-2 is both complex and heterogeneous. Therefore, it is recommended the discussion could include some of the following points to put the study findings into context. (Some of which could also be included in the introduction):

  • Is this the first study to examine this vaccine construct and demonstrate cross-reactive Abs?
  • Are there any other relevant studies in humans or other animal models? Have other vaccines been successful in inducing both IgG and IgA Abs?
  • A discussion re. the importance of / different pathways of production of IgG Abs versus IgA Abs (mucosal-associated). Eg as asymptomatic patients elicit IgG as opposed to IgA or IgM, is it more important to elicit IgG responses?
  • A comparison of the kinetics of IgG and IgA Ab responses (other studies in the literature found IgA to dominate the early neutralising Ab response)
  • A discussion revolved around elicited cross-reactive neutralizing Abs as demonstrate by the results would be interesting
  • A discussion re. the magnitude of Ab responses compared to other vaccines
  • While the authors concentrate primarily on the humoral response, some reflection on the importance of T cell responses would strengthen the paper.
  • The applicability of the mouse model to humans. That is, are the results relevant to humans?
  • The strengths and limitations of the study (eg. small same size of 4-5 nice per group as indicated in figure 2)

Line 270 – Finally the last sentence is flawed. Surely focusing on both accessibility and research surrounding vaccine specificity is important.

Minor points

Page 2, line 51 – An abbreviation for VLP is needed

Page 2, line 52 – include reference for universal TT epitopes (eg. Reece et al, J Immunol 1993

, vol. 151 (pg.6175-84)

Suggest using fewer acronyms to improve the flow of reading the manuscript. Eg., Page 9, line 251 – RBM is not needed.

Author Response

Comment 1: Page 1 – The abstract is a little misleading. It suggests that the cross-reactive IgG Ab experiments were not performed in mice as only the IgA Ab experiments were reported as being conducted in mice (leading to the assumption that these experiments were conducted in another animal model or humans). Suggest clarifying that all experiments were in mice (discovered after reading the methods). Therefore, suggest the statement  ‘VLPs are capable of inducing protective cross-reactive IgG and IgA Abs’ should have “in mice” added.

Reply: We have added “in mice” in revised text.

Comment 2: Page 1, line 35 – need to include a reference to that particular variant enhance affinity for the ACE2 receptor and infectivity of latter strains.

Reply: We have added related references.

Comment 3: Page 2, line 51 – suggest describing / expanding that CuMVTT is a vaccine to the reader who may not be familiar with this field.

Reply: We have added more information about CuMVTT in revised version.

Comment 4: It is not clear how many mice were in each group. It appears from Fig 2 to be 4-5. Suggest outlining sample size of each group in the methods.

Reply: We have added mice number in method 2.5.

Comment 5: Figure 5 - The authors are to be commended for examining the cross-neutralising ability of induced Abs in comparison to just examining seroconversion. However, it would be more appropriate to label variants as alpha, beta, delta, gamma, etc for consistency and clarity.

Reply: We have changed the figure 5 according to your suggestion.

Comment 6: Would the authors consider including a figure comparing the kinetics of IgG versus IgA induction?

Reply: We do not plan to include a figure comparing IgG and IgA antibodies, because the comparison was performed in previous study: Bessa J, Schmitz N, Hinton HJ, Schwarz K, Jegerlehner A, Bachmann MF. Efficient induction of mucosal and systemic immune responses by virus-like particles administered intranasally: implications for vaccine design. Eur J Immunol. 2008 Jan;38(1):114-26. The IgA antibody titers in serum was also lower than IgG titers after Qb-VLP immunization and the IgA response was delayed.

Comment 7: Page 9, line 245 – The discussion (one large paragraph) focuses predominantly on whether it will be necessary to generate yearly versions of the SARS-CoV-2 vaccine. The standard format for a discussion is to provide a  summary of the overall study findings in the first paragraph (there is a only brief outline of the results in point 6 but no specific mention of IgG and IgA), then discuss these results with respect to other study findings and clinical implications of the results.

While the one paragraph is of interest discussing whether it will be necessary to generate yearly versions of the SARS-CoV-2 vaccine, eliciting Ab responses to SARS-CoV-2 is both complex and heterogeneous. Therefore, it is recommended the discussion could include some of the following points to put the study findings into context. (Some of which could also be included in the introduction):

  • Is this the first study to examine this vaccine construct and demonstrate cross-reactive Abs?
  • Are there any other relevant studies in humans or other animal models? Have other vaccines been successful in inducing both IgG and IgA Abs?
  • A discussion re. the importance of / different pathways of production of IgG Abs versus IgA Abs (mucosal-associated). Eg as asymptomatic patients elicit IgG as opposed to IgA or IgM, is it more important to elicit IgG responses?
  • A comparison of the kinetics of IgG and IgA Ab responses (other studies in the literature found IgA to dominate the early neutralising Ab response)
  • A discussion revolved around elicited cross-reactive neutralizing Abs as demonstrate by the results would be interesting
  • A discussion re. the magnitude of Ab responses compared to other vaccines
  • While the authors concentrate primarily on the humoral response, some reflection on the importance of T cell responses would strengthen the paper.
  • The applicability of the mouse model to humans. That is, are the results relevant to humans?
  • The strengths and limitations of the study (eg. small same size of 4-5 nice per group as indicated in figure 2)

Line 270 – Finally the last sentence is flawed. Surely focusing on both accessibility and research surrounding vaccine specificity is important.

Reply: We specifically wanted to have a concise discussion on exactly what we have done. We have written in previous papers about all the aspects raised by the reviewer and it would be very difficult not to repeat ourselves. So we would prefer to leave discussion as it is, to retain the readers attraction and to save space. We changed, however, the last sentence as the reviewer suggested.

Reviewer 4 Report

In this manuscript, Chang and colleagues have shown, using a mouse model, that WT RBD displayed on CuMVTT can be a COVID-19 vaccine candidate for eliciting multivalent protectivity and inducing antibodies to recognize and neutralize multiple VoCs. This study offers a model system possibly advantageous in generating neutralizing antibodies to a wide range of RBD of VoCs and further suppressing those VoCs. This manuscript is well written and some of data are interesting. I have some comments.

  • In Figure 1B: The asterisk in red is shown to indicate CuMVTT-RBDwt conjugation band. How can this band be thought to correspond to the CuMVTT-RBDwt? Did authors confirm it by Western blot? If yes, please show it. Also, the authors need to describe its expected size.
  • In Figure 1C: The authors need to show that these bands correspond to the expected RNA and protein.
  • Lines 176-178, it is written that recombinant RBD proteins were produced. Please provide their quantified yields and concentrations. Were their productions comparable one another? It would be nicer if their Coomassie staining results are shown.
  • Lines 19-20: Please revise some phrases. Suggestion: … not only induce strong…, but also bind RBDs…?
  • Line 40: it’s -> its
  • Line 123: 2.4 was cited. Double check it if 2.4 is correct instead of 2.6.
  • 7 M urea was used to wash the binding and probe avidity. For this approach, if any references were referred to, please cite them in the methods.
  • If there is any negative control available in data of Figure 5, it would be better to include it.

Author Response

Comment 1: In Figure 1B: The asterisk in red is shown to indicate CuMVTT-RBDwt conjugation band. How can this band be thought to correspond to the CuMVTT-RBDwt? Did authors confirm it by Western blot? If yes, please show it. Also, the authors need to describe its expected size.

Reply: We regard the extra band which does not appear in CuMVTT and RBD lanes was the conjugation band. The band is around 56.3kD. This is described now more clearly in the revised legend Figure 1B.

Comment 2: In Figure 1C: The authors need to show that these bands correspond to the expected RNA and protein.

Reply: VLPs migrate in agarose gels as a function of molecular weight of the particle as well as its charge. There is therefore no “expected” size as every VLPs migrate in a unique fashion. The best way to show that we have intact VLPs is the overlay for the RNA stain with the protein stain as intact VLPs are loaded with RNA. This is the case in Figure 1C.

Comment 3: Lines 176-178, it is written that recombinant RBD proteins were produced. Please provide their quantified yields and concentrations. Were their productions comparable one another? It would be nicer if their Coomassie staining results are shown.

Reply: The yields of RBD417, RBD484, and RBD501 were comparable to RBDwt, while RBDtrip was approximately 3-5 times less than RBDwt. The SDS-PAGE gel stained with Coomassie was shown as Figure S1.

Figure S1. SDS-PAGE gel of RBDwt, RBD417, RBD484, RBD501, RBDtrip and RBDdm.

Comment 4: Lines 19-20: Please revise some phrases. Suggestion: … not only induce strong…, but also bind RBDs…?

Reply: We have corrected the text accordingly.

Comment 5: Line 40: it’s -> its

Reply: We have corrected the text accordingly.

Comment 6: Line 123: 2.4 was cited. Double check it if 2.4 is correct instead of 2.6.

Reply: Thank you very much, we have changed 2.4 to 2.6.

Comment 7: 7 M urea was used to wash the binding and probe avidity. For this approach, if any references were referred to, please cite them in the methods.

Reply: We have added the reference in revised manuscript.

Comment 8: If there is any negative control available in data of Figure 5, it would be better to include it.

Reply: We have published the data of mixed CuMVTT and RBDwt elicited mice sera to neutralize Wuhan SARS-CoV-2 strain, which was significantly low (Ref 20). Unfortunately, the ability to neutralize VOCs of these sera was not determined.

Round 2

Reviewer 1 Report

The manuscript quality of the new version is much improved. Therefore, I believe that readers will be interested in this study.

Reviewer 4 Report

The authors have addressed my comments.